# A Survey on the Effect of the Chemical Composition on the Thermal, Physical, Mechanical, and Dynamic Mechanical Thermal Analysis of Three Brazilian Wood Species

**DOI:** 10.3390/polym16182651

**Published:** 2024-09-20

**Authors:** Matheus de Prá Andrade, Heitor Luiz Ornaghi, Francisco Maciel Monticeli, Matheus Poletto, Ademir José Zattera

**Affiliations:** 1Postgraduate Program in Engineering of Processes and Technologies, University of Caxias do Sul, Caxias do Sul 95070-560, Brazil; mpandrade@ucs.br (M.d.P.A.); ornaghijr.heitor@gmail.com (H.L.O.J.); mpolett1@ucs.br (M.P.); ajzatter@ucs.br (A.J.Z.); 2Department of Aerospace Structures and Materials, Faculty of Aerospace Engineering, Delft University of Technology, 2629 HS Delft, The Netherlands

**Keywords:** wood, correlation, thermal properties, mechanical properties

## Abstract

Wood is a versatile material extensively utilized across industries due to its low density, favorable mechanical properties, and environmental benefits. However, despite considerable research, the diversity in species with varying compositions and properties remains insufficiently explored, particularly for native woods. A deeper understanding of these differences is crucial for optimizing their industrial applications. This study investigated the composition, tensile strength, flexural strength, Young’s modulus, bending stiffness and elongation at break, thermal behavior, and viscoelastic properties of three Brazilian native wood species: *Araucaria angustifolia* (ARA), *Dipterix odorata* (DOD), and *Tabeuia ochracea* (TOC). The density of these woods showed a linear correlation with mechanical properties such as Young’s modulus (0.9) and flexural modulus (0.9). The research revealed a linear correlation between the woods’ density and mechanical properties, with lignin content emerging as a key determinant of thermal stability. This study highlights the importance of understanding wood species’ composition and physical properties, and provides valuable insights into their behavior.

## 1. Introduction

Wood remains a highly competitive material today, with one of its key advantages being its beneficial impact on CO_2_ emissions. Wood generally captures more CO_2_ than it emits; for instance, 600 kg of wood can sequester approximately 1.5 tons of CO_2_ [1]. Its composition primarily includes cellulose, hemicellulose, lignin, and low molecular weight components called extractives [2,3,4]. Cellulose is generally responsible for the mechanical properties, while lignin is responsible for thermal stability (the end of the degradation curve) due to its antioxidant capacity and the presence of aromatic rings [3,5,6]. A high ratio of hemicellulose to extractives may be preferred in applications where stiffness and thermal stability (in the initial plateau of the degradation curve) are required, but at the cost of higher moisture absorption, which also acts as an accelerator of thermal degradation [2,3]. Wood’s chemical composition, climate, and region also contribute to the diversity of wood species worldwide. Commonly studied species, such as *Pinus elliot* and *Eucalyptus grandis*, are widely used in industry, resulting in numerous investigations of their mechanical and thermal properties [3]. Figure 1 shows a schematic representation of the cellular structure of wood species [7].

The extensive tropical native forests in Brazil, which account for approximately 12% of the world’s forest area, are home to diverse wood species with considerable variety in their composition and properties. On the other hand, only a limited number of studies have focused on elucidating the properties of these lesser-known species [8], such as their mechanical, physical, and thermal properties. The study of these species is essential for determining their optimal applications and identifying the industries where they can be most effectively utilized. By understanding their mechanical properties, it becomes possible to assess whether their use in specific applications is reliable and suitable for long-term performance. This knowledge also makes it possible to compare the woods’ properties between the wood classes or with other material classes, such as polymers and metals [1].

Wood is a natural composite and an orthotropic material, meaning its properties vary along three distinct directions: longitudinal, radial, and tangential [1]. Additionally, moisture content plays a significant role in affecting its mechanical properties. As the moisture content increases, there is typically a reduction in properties such as the modulus of elasticity. This is because water can act as a plasticizer, weakening the bonds between cellulose fibers and thereby reducing the overall strength of the material [1,9,10].

The mechanical properties of wood are mainly influenced by the density and cellulose content [11,12]. Pertuzzati et al. (2018) [13] studied the effect of mechanical densification on two fast-growing species, pine and eucalyptus. The authors claimed an increase in the mechanical properties with density (from 0.53 to 0.99 g·cm^−3^), with a 294% increase in the modulus of rupture and an 85% increase in Young’s modulus for the pine species. To confirm these results, Li et al. [14] investigated the effect of woods’ properties on the fire performance of glulam columns made of six different wood species in China: poplar, Chinese fir, Douglas fir, hemlock, larch, and spruce. The densities ranged from 0.36 to 0.645 g·cm^−3^, and the fire tests showed lower charring rates and slower heat penetration rates for the higher-density glulam columns.

Yue et al. [15] studied thermally treated Chinese poplar wood, with a focus on structural applications, and claimed that the variation in mechanical properties is directly related to the thermal degradation of the chemical composition. Also, from a structural point of view, the best temperature tested in the treatment of poplar wood was 170–180 °C. Qian et al. [16] increased bamboo’s strength and thermal stability for sustainable construction. The authors produced an environmentally sustainable composite of modified glued bamboo and obtained an improvement in all mechanical properties from 40 to 120%. The composite also showed excellent flame-retardant properties. Pereira et al. [17] investigated the effect of chemical treatment of pineapple crown fiber on the production, chemical composition, crystalline structure, thermal stability, and kinetics of thermal degradation. The authors used different methods to extract cellulose from pineapple crown fiber. The authors claimed that stability, thermal stability, and some physical and chemical properties were improved, depending on the chemical treatment used. Pereira et al. [18] obtained bleached cellulose from orange bagasse using different sequential chemical treatments. Three different pretreatments were used: alkaline treatment, organosolv and residual insoluble alcohol. All pretreated samples were bleached. The FTIR and XRD results indicated that hemicellulose and lignin were largely removed after the first pretreatment. The thermal degradation curves also showed the presence of two degradation steps instead of three. 

Thermal properties are critical, especially when the wood species is incorporated into polymeric matrices [19,20]. To understand the thermal properties of wood and other lignocellulosic materials, the most common technique is thermogravimetric analysis (TGA), from which it is possible to verify the degradation steps, kinetics, and thermodynamic parameters, as many studies have already demonstrated [21,22,23,24,25]. Tabal et al. [26] evaluated the thermal degradation of *Ficus nitida* wood using isoconversion methods such as the Friedman, Flynn–Wall–Ozawa (FWO), and Vyazovkin methods. The authors claimed that the degradation of *Ficus nitida* wood can proceed through three-stage degradation, with activation energies ranging from 171.4 to 248.23 kJ·mol^−1^. Ornaghi Jr. [27] studied the thermal degradation effect of each component of biomass (cellulose, hemicellulose, and lignin, with their respective chars, among others) using a statistical kinetic approach. The main results indicated that cellulose plays a major role in the activation energy, while the reaction order is determined by hemicellulose and lignin. In a review, Hill et al. [28] investigated the effect of chemical modification and hygroscopicity on the thermal properties of various woods. The authors reviewed the current status of thermally modified wood under dry and wet conditions and the role of some polysaccharides in lignocellulosic materials on the thermal properties, dimensional stability, and mass loss. The main information is on the role of OH groups in the sorption behavior and how this affects other properties.

The properties of native woods in Brazil are not well-established, possibly due to the dominance of exotic species in the market, such as *Pinus elliottii* and *Eucalyptus grandis*, which have been the subject of far more research [13,29,30,31]. In this study, three native Brazilian wood species were investigated: *Araucaria angustifolia* (ARA), *Dipteryx odorata* (DOD), and *Tabebuia ochracea* (TOC). *Araucaria angustifolia*, a key species in southern Brazil, is known for its relatively low density and well-balanced mechanical properties [32,33,34]. DOD is present in south of Brazil and spreads over the north and midwest regions; it is known for its high density (>950 kg·m^−3^) and mechanical resistance [35,36]. These species were selected in part because they are native to the southern region of Brazil, making them representative choices for this investigation.

Many other studies of wood properties may focus on the composition, dynamic mechanical thermal analysis, or physical properties, which are generally analyzed without considering other aspects of wood, and correlations among the properties studied are scarce. The aim of this work was to evaluate and correlate the composition and thermal, physical, mechanical, and dynamic properties of three Brazilian wood species: *Araucaria angustifolia* (ARA), *Dipterix odorata* (DOD), and *Tabeuia ochracea* (TOC).

## 2. Materials and Methods

### 2.1. Materials

Three Brazilian native woods were studied in panel format without any prior treatment: *Araucaria angustifolia* (ARA), *Dipterix odorata* (DOD), and *Tabebuia ochracea* (TOC). These wood samples were exclusively extracted from the heartwood, with no portions from the outer sections of the tree utilized. Figure 2 schematically shows the process of obtaining samples for tensile and bending tests and the corresponding characterizations. Wood was obtained from a local provider, Madeireira Bianchi (Bento Gonçalves, RS, Brazil), in board format. Samples for the flexural and tensile tests were machined from the wood board using a computer numerical control (CNC) machine (Jaraguá, SC, Brazil) with the machining aligned along the fibers’ direction. Prior to analysis, the samples were oven-dried at 105 °C for 4 h to eliminate residual moisture.

### 2.2. Composition

The composition of three wood samples was measured using established methods. Wood extractives were determined according to TAPPI T204 [37] using a 1:2 ethanol/benzene solution. The Klason lignin content was determined according to TAPPI T222 [38]. The Van Soest method was used for factors such as moisture, cellulose, and hemicellulose. All tests were performed in triplicate to ensure the reliability of the results.

### 2.3. Mechanical Tests and Density

Two tests were used to measure the mechanical properties of wood. The bending tests were carried out on an EMIC (Caxias do Sul, Brazil) universal testing machine, model DL 3000 (Instron, Norwood, MA, USA), with a 200 kg load cell. The tests were carried out following ASTM D790 [39], with rectangular samples measuring 125 × 12.7 × 3.2 mm. The tensile tests were performed again on an EMIC DL 3000 (Caxias do Sul, RS, Brazil) universal testing machine under ASTM D638 [40], with base dimension of 165 × 13 × 3.2 mm. The standards were used without any adjustment.

In order to understand the relationship between the physical and mechanical properties, the densities of the wood species were also measured. This was determined by the volumetric method according to ASTM D792 [41], using the machined specimens from the bending tests (25 × 13 × 4 mm), and the samples were oven dried for 4 h at the temperature of 105 °C. Equation 1 was used to determine the density of the wood.
(1)$$\rho =\frac{\left(a*b\right)}{\left(a-c\right)}$$
where *ρ* is the density (g·cm^−3^), *a* is the sample’s mass (g), *b* is the density of water (g·cm^−3^), and *c* is the sample’s mass underwater (g·cm^−3^). Five samples were used for each of the three wood species for each measurement.

### 2.4. Statistical Analysis

The coefficient of variation (CV) is a statistical measure that represents the ratio of the standard deviation to the mean of a dataset, which represents the standard deviation divided by the average (Equation (2)). It is expressed as a percentage and used to compare the relative variability of data between different datasets or measures. A higher CV indicates greater variability relative to the mean, while a lower value indicates greater consistency.
(2)$$CV\;\left(\%\right)=100\cdot \frac{\sigma }{\mu }$$
where *σ* is the standard deviation and *μ* represents the average.

A Weibull model (Equations (3) and (4)) was used to assess the reliability of the mechanical test data. In this model, the variable *x* represents the measured physical and mechanical properties, i.e., density, tensile strength, Young’s modulus, elongation at breaking, flexural strength, and modulus; *β* represents the shape parameter indicating whether the distribution follows an exponential trend (*β* = 1) or a polynomial trend (*β* > 1). The *α* is the scale parameter, reflecting the scale of the measured values. The function *F*(*x*) is the probability density function describing the probability of each variable taking a particular value, which is directly related to the confidence level, where *R*(*x*) = 1 − *F*(*x*). OriginLab 2021 software was used to fit the reliability curves for each parameter. The purpose of this method was to activate the 95% reliability level for all physical and mechanical parameters.
(3)$$F\left(x\right)=1-exp{\left[-(x/\alpha )\right]}^{\beta }$$
(4)$$ln\left\{ln\left[1/R(x)\right]\right\}=\beta \cdot ln(x)+\beta \cdot ln(\alpha )$$

### 2.5. Thermal Analysis

The samples were subjected to the thermogravimetric analysis (TGA) for thermal analysis. TGA was performed using Shimadzu (Kyoto, Japan) TGA-50 equipment, using a nitrogen atmosphere with a flow of 50 mL·min^−1^, starting from 25 up to 800 °C, a heating rate of 10 °C·min^−1^, and a platinum crucible. Prior to testing, the samples were oven-dried at 105 °C for 4 h to minimize the residual moisture content.

### 2.6. Dynamic Mechanical Thermal Analysis (DMTA)

All samples were subjected to DMTA in a dynamic mechanical analyzer from TA Instruments (Newcastle, WA, USA), Model Q800, with a dual cantilever mode. The storage modulus (E′), loss modulus (E″), and damping factor (tan δ) were obtained. The analysis was conducted from −130 °C to 150 °C with a heating rate of 5 °C·min^−1^. The deformation was set at 0.1%, and the frequency was 1 Hz. The samples were dried in an oven at 105 °C for 4 h before testing to reduce any remaining moisture content.

## 3. Results and Discussion

### 3.1. Composition

The composition of the materials tested is shown in Table 1. This table shows the average results of three measurements for each sample and the standard deviation. Cellulose presented lower variation and lower CV as a consequence. In terms of the results, it can be seen that for constituents with a higher proportion, there was less variability due to the greater sensitivity of the measurements and the reliability of their values, whereas for those with reduced proportions, there was less sensitivity and greater variability in the results.

ARA and DOD had a higher cellulose content than TOC. In theory, the higher the cellulose content, the better the mechanical properties, because cellulose is made up of long, linear chains of glucose molecules linked together by strong hydrogen bonds, forming very stable and rigid structures that can improve the tensile strength of the wood. Another important aspect of these materials is their lignin content. TOC had the highest lignin content of the woods studied, reaching a total of 48.9%. Lignin is an amorphous biopolymer with a very complex structure compared with the other components. With its high molecular weight and aromatic rings, lignin is a natural material with high thermal stability, decomposing over a wide temperature range (200 °C to 500 °C). However, the degradation of some components of lignin can also occur at temperatures below 200 °C. Notably, lignin is often selected as a natural antioxidant in polymer applications, as reported by several authors [42,43,44,45]. DOD showed the highest extractive content, reaching 5.33%, but was similar to TOC, considering the standard deviation. The higher the concentration of extractives, the lower the thermal stability. The degradation of low molecular weight compounds accelerates the thermal degradation of the material. Finally, the ash content of all samples ranged from 0.28% to 0.33%.

The hemicellulose content was very similar for all woods, with ARA and DOD showing almost identical results of 9.1% and 9.7%, respectively. DOD showed a slightly lower hemicellulose content of 7.4%. Hemicellulose is considered to be the primary substance in wood cell walls, as it is the material that coats and shapes the cellulose structure and is responsible for the flexibility of lignocellulosic materials. Therefore, wood with a high hemicellulose content can have a more flexible structure and absorb more moisture. However, hemicellulose can act as an accelerator of degradation, reducing the thermal stability of these materials [2,46,47].

The results found for ARA were consistent with the literature, where Barros et al. (2021) [48] found that *Araucaria angustifolia* contained a total of 7% extractives, 34% lignin, 9% hemicellulose, and 46% α-cellulose.

### 3.2. Mechanical Tests and Density

The wood samples were subjected to mechanical tensile and bending tests, and density measurements. The results are presented in Table 2, where the average results of five sample measurements are shown.

Considering that the variability of each result was high, Weibull methodology was applied to activate the values, with a reliability of 95%. The Weibull equation followed the procedure in [49]. The results presented in Figure 3 represent the values found at a 95% confidence level. In general, they were lower than the averages due to the influence of the variability of the replicates, in addition to the high level of rigor applied. The averages presented a reliability close to 60%, which was less conservative. However, the trends were the same as those observed for the average.

TOC showed the highest values for tensile properties, reaching 69.30 MPa for tensile strength and 1.27 GPa for Young’s modulus. DOD also showed high values for both properties, with a very similar value for Young’s modulus. In the flexural tests, ARA still showed the lowest values of the species analyzed, with the most notable result being the flexural strength, which was 45% lower than DOD. These properties were probably most affected by the density of these materials. Table 3 shows that DOD and TOC had significantly higher densities than ASU, with 1.02 g·cm^−3^ and 0.88 g·cm^−3^, respectively. In general, the cellulose content is critical in terms of the mechanical properties. However, TOC outperformed DOD in some of these properties, even though DOD had a higher cellulose content. 

The most notable difference was flexural strength, where DOD had a value of 223.81 MPa compared with ARA, which had 136.09 MPa, indicating a more rigid behavior. It can be observed that density has a more significant effect on mechanical performance than cellulose content in the case of the properties of TOC and ARA. Although ARA had a higher cellulose content, the density of TOC was almost twice that of ARA, suggesting a significant influence on the mechanical properties. The oven-dry density of ARA was up to twice that of DOD. Correspondingly, the flexural strength of ARA followed this trend and was significantly higher than that of DOD. However, the tensile strength of both materials was almost identical. This study did not focus on the fibers’ orientation or grain structure, but these factors may account for the observed differences in the mechanical properties. In addition, the presence and variability of defects may have played a significant role [50,51]. In particular, the tensile strength of DOD had twice the standard deviation of ARA, suggesting that the variability of the DOD samples may have been greater than those of ARA. 

The Pearson correlations between some of the woods’ composition and mechanical properties is shown in Figure 4. Three wood samples were used to calculate the correlations among these properties.

Pearson’s correlation coefficient measures the linear correlation between two variables and helps to identify variations in the woods’ composition correlated with the mechanical properties. A correlation coefficient close to −1 or 1 indicates a high linear correlation between two variables, whereas a value close to −1 indicates a negative correlation. The graphic shows the correlation of each mechanical property with the thermal and other properties. The heat map shows a clear correlation between the density and mechanical properties of wood, particularly Young’s modulus (0.9), flexural strength (0.8), and flexural modulus (0.8). In addition, hemicellulose showed a negative correlation with most of the mechanical properties, suggesting that hemicellulose plays a negative role in wood’s properties when it comes to the mechanical properties investigated in our study (tensile strength, flexural strength, modulus of elasticity, bending stiffness, and elongation at breaking).

In contrast, in a study by Yu et al. [15], the authors claimed that hemicellulose determined the longitudinal shear strength of wood. This effect could be attributed to the chemical properties of hemicellulose. As a heteropolymer composed of sugar monomers that form a matrix around cellulose microfibrils, hemicellulose can act as a plasticizer, especially at higher temperatures [52]. 

### 3.3. Thermal Analysis

Thermal analysis was performed on all samples, and the TGA and differential thermogravimetric analysis (DTG) curves are shown in Figure 5.

Figure 5 shows the TGA and DTG of ARA samples, and three main degradation steps can be identified. An initial mass loss was observed around 100 °C, associated with releasing chemically bound water and some low molecular weight components. The first degradation step was observed around 200 °C and was mainly attributed to hemicellulose. Hemicellulose, being the component with the lowest thermal stability among the main elements of wood, contributed to this degradation step [2]. As a comparison of the thermal stability among the species, the temperature at which 10% mass loss occurred (T10) was measured at a heating rate of 10 °C·min^−1^. For ARA, T10 was 275.5 °C and the temperature at which the sample reached its maximum degradation rate (Tmax) was 348.4 °C. 

A second stage occurred at around 370 °C and was closely linked to the first. This stage was mainly concerned with the degradation of cellulose. The thermal degradation of cellulose started in the amorphous phase and progressed to the crystalline phase as the temperature rose, due to its more stable structure. In the case of ARA, a third step was unclear, but it is very common for wood materials to present a third degradation step related to lignin. Lignin undergoes thermal degradation over a wide temperature range, initiating the degradation of less stable structures. Due to its complex structure of benzene–propane units, which are highly cross-linked and have a high molecular weight, lignin requires significantly more energy to degrade than hemicellulose and cellulose. Consequently, the degradation of lignin continues until the wood sample is entirely degraded.

Figure 5 shows the TGA and DTG for DOD. TGA clearly showed three degradation steps for DOD compared with ARA, a behavior that is more commonly observed in wood species. T10 occurred at 285.4 °C and Tmax at 360 °C. The first step, which started at around 200 °C, involved the degradation of low molecular weight compounds such as extractives and hemicelluloses. In the case of DOD, the degradation of hemicelluloses could be verified by DTG, where a shoulder was observed in the range of 210–315 °C. In the second stage, starting at around 320 °C and continuing up to 400 °C, cellulose degradation took place. Starting at 400 °C, the third stage is generally associated with lignin degradation. DOD showed a slightly higher thermal stability compared with ARA, with a shift in T10 and a slightly more pronounced difference in Tmax, with a variation of 12 °C. Although the composition of these two species was very similar, DOD had a higher lignin content, suggesting that the antioxidant behavior of lignin may have improved its thermal stability. However, the main difference between these species may be their density. As the density of DOD was twice that of ARA, its more compact cell structure may make the diffusion of degradation more difficult. Another relevant aspect is that ARA showed more pronounced water retention, although the samples were subjected to the same drying process before TGA. The water content may have facilitated the initiation of degradation, potentially reducing the thermal stability [27,53]. 

Similar to what was observed for ARA, the TGA curve of TOC in Figure 5 shows a mass loss below 100 °C, which is related to residual moisture that was not completely removed during the drying process. The general behavior observed was the same as that of DOD. For TOC, T10 occurred at 288.2 °C and Tmax at 370 °C. The first degradation step started at around 212 °C, where a shoulder attributed to hemicellulose could be observed for DTG. The second degradation step began at around 326 °C and extended to 415 °C, where the third and final step occurred. Compared with ARA and DOD, all temperatures observed for TOC were higher, indicating a higher thermal stability compared with ARA and DOD. 

In brief, the three main components of wood fiber are hemicellulose, cellulose, and lignin. Hemicellulose consists of xylose, mannose, glucose, galactose, and other saccharides in an amorphous and branched structure. Cellulose is a glucose chain without branches and has higher thermal stability than hemicellulose. This property is achieved because cellulose has a higher chain order, resulting in a higher packing of the glucose chains. Finally, lignin also has branches with aromatic rings, resulting in degradation over a wide temperature range. In other words, hemicellulose degrades first (200–320 °C), followed by cellulose (320–400 °C) and lignin (150–900 °C). It is worth noting that the degradation events overlap at specific temperatures. Chemically, hemicellulose has a higher CO and CO2 yield, while lignin has a higher CH4 yield. The organic compounds are mainly released at lower temperatures (<500 °C) from hemicellulose and cellulose. Lignin has a low release of organic compounds below this temperature [54,55].

Several aspects of wood and lignocellulosic materials contribute to their thermal stability. The mass fraction ratio of the components plays an important role, with higher ratios of extractives and hemicelluloses leading to earlier degradation. In addition, the complex structure of lignin requires more energy to degrade. Therefore, if a lignocellulosic material has a higher lignin content, it may also have a higher thermal stability [27,53,56]. In the case of TOC, it had the highest lignin content, combined with its relatively high density. These properties likely contributed significantly to the observed thermal stability of the TGA results.

### 3.4. Dynamic Mechanical Thermal Analysis (DMTA)

The DMTA of the wood samples is presented in Figure 6. This technique was employed to investigate the viscoelastic properties of the studied woods. The graphic shows the storage (E′) (Figure 6a) and loss modulus (E″) (Figure 6b), which represent the elastic and viscous responses, respectively.

Within the loss modulus (Figure 6b), three relaxation events could be observed for each wood sample. The first observed event was visualized in the range of −95 to −89 °C. The specific temperatures are given in Table 3 and are labelled γ. This third-order relaxation could be attributed to the methylol and hydroxymethyl groups in non-crystalline regions of hemicellulose or cellulose, which agrees with the observation of several authors [31,56,57]. 

At a higher temperature range, from −23.9 to 12.9 °C, the β relaxation, also a secondary event, was observed. The interpretation and location of the peaks for this event can be challenging, as they may overlap with the γ relaxation event. This event can be attributed to the loss of hemicellulose or the moisture content in the hemicellulose regions, as suggested by previous studies [57]. The observation of this peak may be more complicated, depending on the moisture content of the wood [31].

The last set of peaks, observed above 100 °C, can be related to the micro-Brownian motion of the polymer chains during the glassy to viscous transition. This phenomenon is generally associated with the glass transition (T_g_) of lignin [56]. This was only detectable via DMTA because lignin and hemicellulose are primarily amorphous polymers and behave as thermoplastic polymers. Although cellulose is a semi-crystalline material, detecting the relaxation of hemicellulose and lignin is much easier than that of cellulose, and DMTA plays a crucial role in investigating the viscoelastic behavior of wood. 

Only lignin T_g_ could be identified in the temperature range studied, as hemicellulose and amorphous cellulose showed relaxation at higher temperatures, above 200 °C. In Table 3, the α-relaxation was shifted to higher temperatures for DOD and TOC compared with ARA. This shift can be attributed to the higher lignin content in these two species, and it is possible that this result is correlated with the thermal behavior of the wood species. This finding may explain why these species exhibited higher thermal stability, as the lignin-softening temperature was higher, requiring more energy than the lignin T_g_ of ARA. 

Figure 6a plots the storage modulus (E′), showing the differences between the three wood species analyzed, with DOD showing the highest stiffness. The numerical results are tabulated in Table 4, and DOD presented an E′ at 25 °C of 30.16 GPa, which was 44% and 55% higher than that of ARA and TOC, respectively. This result was consistent with the mechanical results, especially the flexural strength and modulus, where DOD showed the highest values compared with ARA and TOC. From the curve, it can be seen that the E values decreased with temperature. This was due to the increase in chains’ mobility in the amorphous phase of the wood constituents [58].

The loss factor (tan δ) was also higher for DOD, as seen in Table 4 and Figure 7, indicating a more dissipative and elastic behavior, consistent with the observed E′ results [56]. 

The effect of wood’s fiber components, hemicellulose, cellulose, and lignin is that they are more difficult to separate in dynamic mechanical thermal analysis. Different chemical and physical interactions in the wood fibers can affect the dynamic mechanical properties. In addition to the intrinsic properties of cellulose, hemicellulose, and lignin, other interactions can affect mechanical resistance, such as ester bonds to hemicellulose for ferulic and p-coumaric acid, ester and ether bonds between p-coumaric acid and lignin, ester and ether bonds between hemicellulose and lignin, hemicellulose cross-linking by diferulic acid, diferulic acid cross-linking (ester to polysaccharides and ether to lignin), and ferulic acid cross-linking (ester to polysaccharides and ether to lignin), among others. Knots, cross-grains, checks, splits, moisture, soil conditions, and growing space also affect the properties of wood [55].

## 4. Conclusions

The study investigated the composition, mechanical properties, thermal behavior, and viscoelastic properties of three Brazilian wood species: *Araucaria angustifolia* (ARA), *Dipterix odorata* (DOD), and *Tabeuia ochracea* (TOC). The species analyzed showed different compositions, with ARA and DOD having a higher cellulose content compared with TOC, while TOC had a higher lignin content. Although wood with a higher cellulose content typically has higher mechanical strength, this was not the case in this study. TOC presented the lower limit of cellulose content and exhibited higher strength and modulus than ARA, suggesting that wood density plays a critical role in mechanical properties, a conclusion supported by Pearson analysis. Thermal analysis indicated that ARA may undergo a two-step degradation process, whereas DOD and TOC undergo a three-step process. TOC showed the highest thermal stability due to its high lignin content and density.

Regarding the viscoelastic properties, DOD showed the highest E′ and tan δ. Relaxation temperatures α, β, and γ were observed. The γ relaxations ranged from −95 to −89, β relaxations were from −23.9 to 12.9, and α relaxations were from 103.7 to 135.9. The storage modulus values that were corroborated by the observed mechanical properties were for DOD, which had the highest values for both. This study highlights the importance of understanding wood’s composition and physical properties when assessing its mechanical and thermal properties. Density and composition can have a direct effect on these properties. Thus, this research contributes to selecting and using wood materials in various applications.

## Figures and Tables

**Figure 1 polymers-16-02651-f001:**
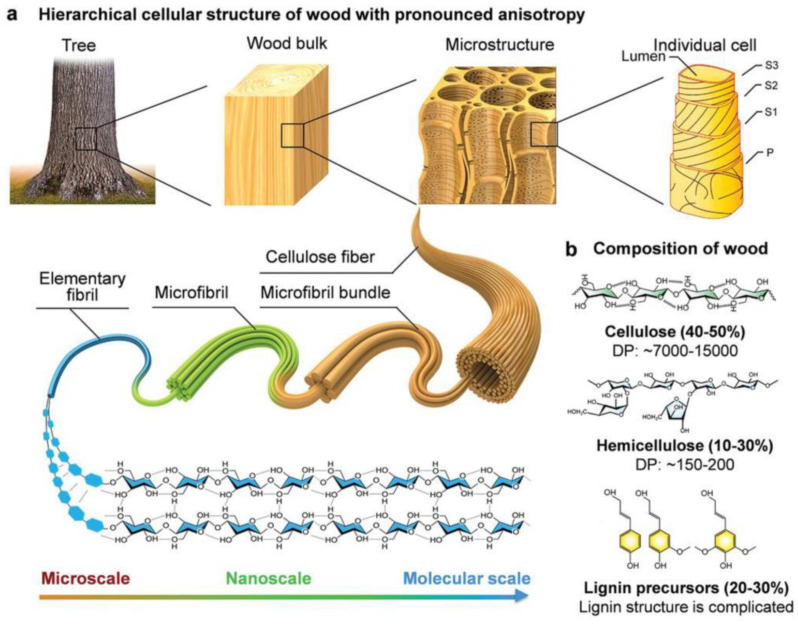
Schematic representation of the cellular structure of wood species showing the (**a**) Hierarchical cellular structure of wood and (**b**) composition of wood (figure used under the terms and conditions of creative commons attribution [7]).

**Figure 2 polymers-16-02651-f002:**
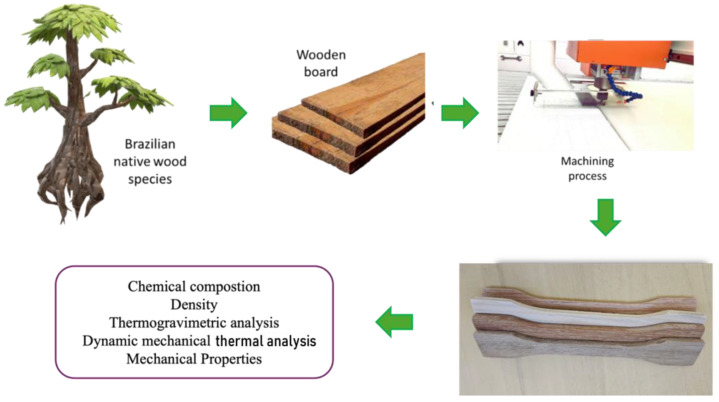
Schematic representation of the process of obtaining the specimens and their characterization.

**Figure 3 polymers-16-02651-f003:**
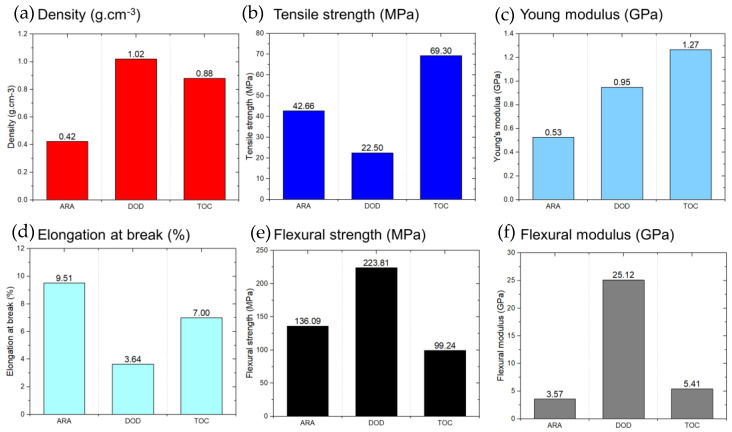
Weibull method reliability (95%) for (**a**) density, (**b**) tensile strength, (**c**) Young modulus, (**d**) elongation at break, (**e**) flexural strength, and (**f**) flexural modulus for different wood species.

**Figure 4 polymers-16-02651-f004:**
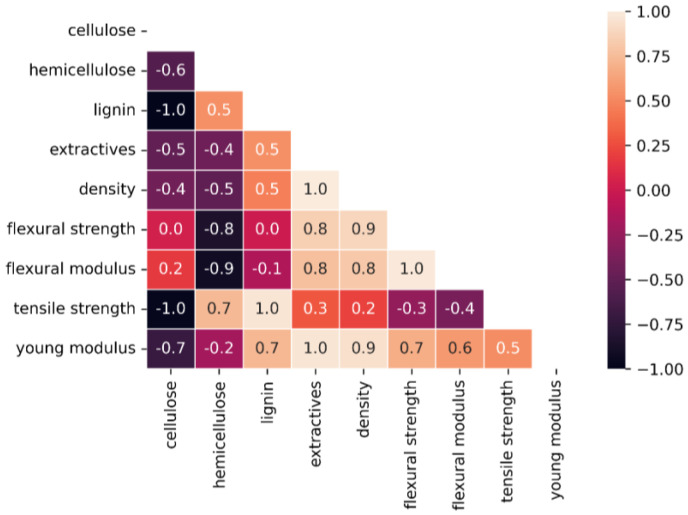
Heatmap of Pearson’s correlations for woods’ composition and mechanical properties.

**Figure 5 polymers-16-02651-f005:**
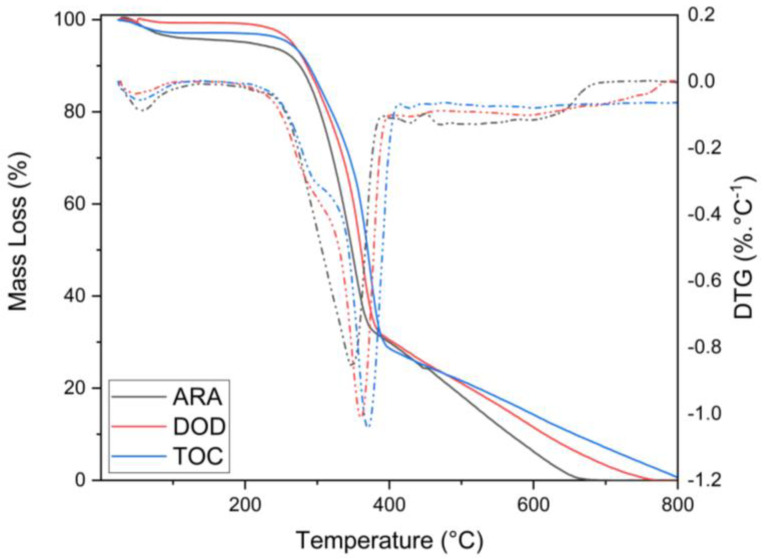
TGA (full lines) and DTG (dashed lines) curves for ARA (black lines), DOD (red lines), and TOC (blue lines).

**Figure 6 polymers-16-02651-f006:**
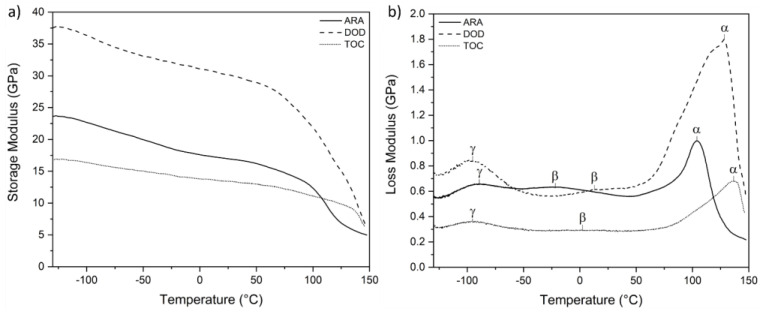
Storage (**a**) and loss modulus (**b**) for the three wood species showing the main thermal transitions.

**Figure 7 polymers-16-02651-f007:**
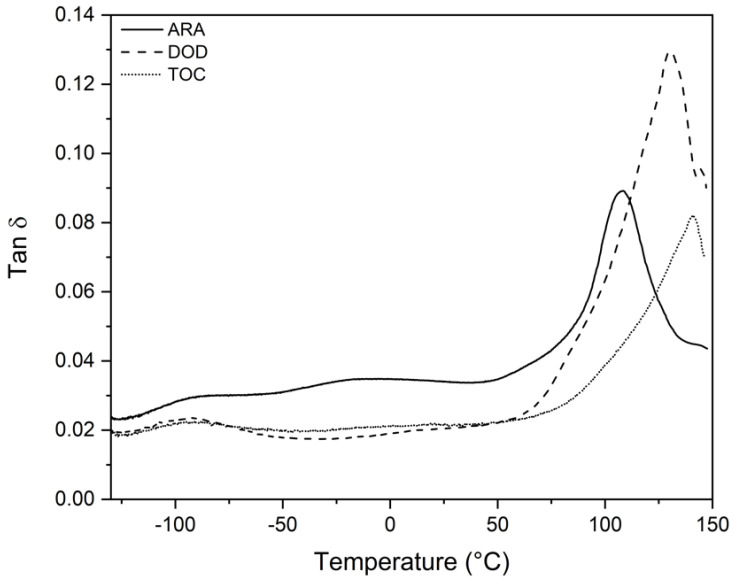
Tan delta for the three wood species.

**Table 1 polymers-16-02651-t001:** Chemical composition of the three wood species.

Sample	Cellulose (%)	*CV (%)	Hemicellulose (%)	*CV (%)	Lignin (%)	*CV (%)	Extractives (%)	*CV (%)	Ash (%)	*CV (%)
ARA	49.9 ± 0.3	0.7	9.1 ± 1.8	20.2	26.8 ± 3.3	12.4	3.48 ± 1.0	28.0	0.33 ± 0.1	35.8
DOD	45.9 ± 3.3	7.1	7.4 ± 0.1	1.3	31.8 ± 2.5	8.0	5.33 ± 0.6	11.3	0.30 ± 0.1	6.9
TOC	27.9 ± 2.9	9.4	9.7 ± 2.3	23.8	48.9 ± 0.9	1.9	4.98 ± 0.1	8.6	0.28 ± 0.1	15.1

*CV, coefficient of variation.

**Table 2 polymers-16-02651-t002:** Tensile, flexural, and density properties of wood species.

Sample	Density (g·cm^−3^)	Tensile Strength (MPa)	Young’s Modulus (GPa)	Elongation at Breaking (%)	Flexural Strength (MPa)	Flexural Modulus (GPa)
ARA	0.486 ± 0.031	67.67 ± 14.6	0.63 ± 0.5	12.9 ± 1.9	134.0 ± 18.7	9.85 ± 3.7
DOD	1.059 ± 0.017	65.05 ± 31.7	1.49 ± 0.3	6.8 ± 1.8	294.6 ± 38.5	30.46 ± 2.8
TOC	0.906 ± 0.012	112.70 ± 32.2	1.57 ± 0.2	10.0 ± 2.1	184.0 ± 25.4	13.64 ± 5.4

**Table 3 polymers-16-02651-t003:** Relaxation temperatures for ARA, DOD, and TOC woods.

	Molecular Transitions		
Sample	γ (°C)	β (°C)	α (°C)
ARA	−89.3	−23.9	103.7
DOD	−92.7	12.9	128.1
TOC	−95.0	2.2	135.9

**Table 4 polymers-16-02651-t004:** E′, E″, and tan δ measurements for ARA, DOD, and TOC woods.

Sample	E′ at 25 °C (GPa)	E″ at 25 °C (GPa)	Tan δ at 25 °C	Tan δ Peak Height	E′ at 25 °C (GPa)
ARA	16.97	0.57	0.034	0.089	16.97
DOD	30.16	0.62	0.020	0.130	30.16
TOC	13.43	0.29	0.022	0.082	13.43

## Data Availability

The original contributions presented in the study are included in the article, further inquiries can be directed to the corresponding author.

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
