# Peer review of "A Survey on the Effect of the Chemical Composition on the Thermal, Physical, Mechanical, and Dynamic Mechanical Thermal Analysis of Three Brazilian Wood Species"

_polymers, 2024, doi:10.3390/polym16182651_

Round 1

Reviewer 1 Report

Comments and Suggestions for Authors

The paper by Matheus de Prá Andrade is devoted to the study of the physicochemical properties of three Brazilian woods. Using thermogravimetric analysis, the thermal stability of the samples was evaluated. Using dynamic mechanical thermal analysis, the α, δ, γ relaxation temperatures were determined. The γ relaxations ranged from -95 °C to -89 °C, β from -23.9 °C to 12.9 °C and α from 103.7 °C to 135.9 °C.

However, the manuscript needs some major revision in terms of writing. My specific comments are listed below.

1. At the beginning of the Introduction section, the actuality of the study of the thermal and mechanical properties of wood and their practical applications is not well understood. The relevance and novelty of the study needs to be completely rewritten and expanded.

2. It is not clear why the following woods were used: Araucaha angustifolia, Dipterix odorata, Tabeuia ochracea. The introduction should be supplemented with specific information where these woods are used.

3. Line 83-87. The introduction discusses different methods for determining the activation energy of the thermal degradation process, but the paper misses a kinetic study, perhaps better to show recent advances in the mechanical analysis of wood.

4. Specify the type and type of crucibles used in the TGA method. Why was a heating rate of 50 K/min used? Due to high heating rates, there is a temperature lag between the sample and the thermocouple, because of the high thermal resistance, also it favours the superposition of several effects on each other, and on the TG curve it is presented as a single stage. 

5. Line 174-176, a reference to this study should be given.

6. As discussed in the Thermal analysis section there is a small amount of water present in the composition of dried woods. It is well known that the glass transition temperature and in general alpha relaxation depends on the amount of water in the sample. In this case, is it possible that the observed mixing of alpha relaxation is due to the presence of water in the samples? Particularly in the ARA sample.  The DOD and TOC wood samples have the same amount of lignin considering the determination error, but the temperature of the alpha relaxation maximum is very different.

7. Line 373. Change δ to γ.

8. Correct references throughout the text (Error! Reference source not found).

9. Line 174-176, a reference to this study should be given.

10. As discussed in the Thermal analysis section there is a small amount of water present in the composition of dried woods. It is well known that the glass transition temperature and in general alpha relaxation depends on the amount of water in the sample. In this case, is it possible that the observed mixing of alpha relaxation is due to the presence of water in the samples? Particularly in the ARA sample.  The DOD and TOC wood samples have the same amount of lignin considering the determination error, but the temperature of the alpha relaxation maximum is very different.

11. Line 373. Change δ to γ.

12. Correct references throughout the text (Error! Reference source not found).

For a better study of the thermal properties of the woods studied, it would be desirable to carry out kinetic analyses (Friedman analysis, CAS method, etc.) of thermal and thermo-oxidative stability. This would allow the optimum performance characteristics of the woods to be further determined. 

Author Response

Comments and Suggestions for Authors

The paper by Matheus de Prá Andrade is devoted to the study of the physicochemical properties of three Brazilian woods. Using thermogravimetric analysis, the thermal stability of the samples was evaluated. Using dynamic mechanical thermal analysis, the α, δ, γ relaxation temperatures were determined. The γ relaxations ranged from -95 °C to -89 °C, β from -23.9 °C to 12.9 °C and α from 103.7 °C to 135.9 °C.

However, the manuscript needs some major revision in terms of writing. My specific comments are listed below.

Comment: At the beginning of the Introduction section, the actuality of the study of the thermal and mechanical properties of wood and their practical applications is not well understood. The relevance and novelty of the study needs to be completely rewritten and expanded.

Answer: The introduction section was expanded and the required information were included.

Comment: It is not clear why the following woods were used: Araucaha angustifolia, Dipterix odorata, Tabeuia ochracea. The introduction should be supplemented with specific information where these woods are used.

Answer: A paragraph explaining the reason was added to the manuscript.

Comment: Line 83-87. The introduction discusses different methods for determining the activation energy of the thermal degradation process, but the paper misses a kinetic study, perhaps better to show recent advances in the mechanical analysis of wood.

Answer: That is correct. We discuss kinetics to highlight the significance and potential of thermal analysis, even though it is not utilized in this paper, but may be important in the general context. We have added additional paragraphs to elaborate on the mechanical properties.

Comment: Specify the type and type of crucibles used in the TGA method. Why was a heating rate of 50 K/min used? Due to high heating rates, there is a temperature lag between the sample and the thermocouple, because of the high thermal resistance, also it favours the superposition of several effects on each other, and on the TG curve it is presented as a single stage. 

Answer: The heating rate was wrong in the paper, the correct value used was 10 C.min-1 and was already updated in the paper. The crucible type was added to the paper.

Comment: Line 174-176, a reference to this study should be given.

Answer: This study reference was added.

Comment: As discussed in the Thermal analysis section there is a small amount of water present in the composition of dried woods. It is well known that the glass transition temperature and in general alpha relaxation depends on the amount of water in the sample. In this case, is it possible that the observed mixing of alpha relaxation is due to the presence of water in the samples? Particularly in the ARA sample.  The DOD and TOC wood samples have the same amount of lignin considering the determination error, but the temperature of the alpha relaxation maximum is very different.

Answer: Yes, moisture content can influence alpha relaxation by affecting the softening point of lignin. While we cannot definitively confirm this, it remains a plausible possibility.

Comment: Line 373. Change δ to γ.

Answer: It was changed.

Comment: Correct references throughout the text (Error! Reference source not found).

Answer: All the broken references were corrected.

Comment: Line 174-176, a reference to this study should be given.

Answer: This study reference was added.

Comment: As discussed in the Thermal analysis section there is a small amount of water present in the composition of dried woods. It is well known that the glass transition temperature and in general alpha relaxation depends on the amount of water in the sample. In this case, is it possible that the observed mixing of alpha relaxation is due to the presence of water in the samples? Particularly in the ARA sample.  The DOD and TOC wood samples have the same amount of lignin considering the determination error, but the temperature of the alpha relaxation maximum is very different.

Answer: Yes, moisture content can influence alpha relaxation by affecting the softening point of lignin. While we cannot definitively confirm this, it remains a plausible possibility.

Comment: Line 373. Change δ to γ.

Answer: This line was changed!

Comment: Correct references throughout the text (Error! Reference source not found).

Answer:All references were corrected.

Comment: For a better study of the thermal properties of the woods studied, it would be desirable to carry out kinetic analyses (Friedman analysis, CAS method, etc.) of thermal and thermo-oxidative stability. This would allow the optimum performance characteristics of the woods to be further determined. 

Answer: We appreciate the comments but the kinetic analysis will be addressed in a future paper. In this study, we have chosen to focus on other properties.

Reviewer 2 Report

Comments and Suggestions for Authors

The manuscript titled ‘A survey on the effect of the chemical composition on the thermal, physical, mechanical, and dynamic mechanical thermal analysis of three Brazilian wood species’ is submitted for probable publication in the journal Polymers. This manuscript deals with some physical and mechanical properties of three Brazilian species using established methods. In my opinion, current research provides important knowledge on these three selected wood species, which were obtained through systematic and scientific analyses, and could thus be beneficial to the scientific community. However, before accepting, this manuscript needs a considerable amount of revision. Please see my comments below:

Abstract: Too much background information. Can be written in one or two lines. After that start from line 19. Scientific/Latin names should be in italic form. As for example: Araucaria angustifolia.

Introduction: Please try to focus on why these three lesser-known species are important. Their habitat, forest coverage, and commercial values. A detailed description of this topic need to be intensively described in this section, especially before the last paragraph.

Line 41: please correct ‘Error! Reference source not found.’ This is found throughout the manuscript where you wanted to cite your figures and tables.

Line 84 and 86: Please write scientific/Latin name properly.

Materials and methods: Need to provide detailed information. Otherwise, difficult to follow.

Line 106: Please provide the source of the material. From which part of the tree were samples obtained? Does it contain any heartwood or juvenile wood? Please provide information in detail. In addition, here you mention the full species name along with the authors. As for example: Dipterix odorata (Aubl.) Willd.

Line 113: Chemical composition

The full references for the TAPPI standards are missing from the reference list. The same applies to the ASTM standards.

Line 119: In what conditions were the samples stored, and for how long? What type of density have you measured- green, oven-dried, or basic density? Please provide the manufacturer, as well as the city and country of production, for the universal testing machine. Even though you mentioned that you followed a standard, it is important to provide the sample size, temperature, and relative humidity at which samples were acclimatized etc.

Line 133: Please mention the amount of samples taken for this analysis. Same comment for the equipment as mentioned above.

Statistical analysis: This section is needed to describe the statistical differences of measured properties among three studied species. Otherwise, it is not scientific to say one species has higher let’s say cellulose than other. Tables 1 and 2 need statistical test data.

Results: Is this section for results only, or does it include both results and discussion? The numbering of the sections is also incorrect, as it jumps from 3. Results to 5. Conclusion.

Line 175, 232, 252, 270: Scientific names should be in italic. Please be consistent throughout the manuscript.

Line 228: Please provide the full meaning of DTG once it appears first.

Line 232-278: This description can be condensed by summarizing Figure 4 instead of describing it separately for each of the three species. Choose one key point and compare it across all three species. Please also support your statements with references from previous findings.

Author Response

Comments and Suggestions for Authors

The manuscript titled ‘A survey on the effect of the chemical composition on the thermal, physical, mechanical, and dynamic mechanical thermal analysis of three Brazilian wood species’ is submitted for probable publication in the journal Polymers. This manuscript deals with some physical and mechanical properties of three Brazilian species using established methods. In my opinion, current research provides important knowledge on these three selected wood species, which were obtained through systematic and scientific analyses, and could thus be beneficial to the scientific community. However, before accepting, this manuscript needs a considerable amount of revision. Please see my comments below:

Comment: Abstract: Too much background information. Can be written in one or two lines. After that start from line 19. Scientific/Latin names should be in italic form. As for example: Araucaria angustifolia.

Answer: Background information was reduced and the scientific names were corrected.

Comment: Introduction: Please try to focus on why these three lesser-known species are important. Their habitat, forest coverage, and commercial values. A detailed description of this topic need to be intensively described in this section, especially before the last paragraph.

Answer: A paragraph was added to explain the importance of those species.

Comment: Line 41: please correct ‘Error! Reference source not found.’ This is found throughout the manuscript where you wanted to cite your figures and tables.

Answer: All the broken references were corrected.

Comment: Line 84 and 86: Please write scientific/Latin name properly.

Answer: The names were corrected.

Comment:  Materials and methods: Need to provide detailed information. Otherwise, difficult to follow. Line 106: Please provide the source of the material. From which part of the tree were samples obtained? Does it contain any heartwood or juvenile wood? Please provide information in detail. In addition, here you mention the full species name along with the authors. As for example: Dipterix odorata (Aubl.) Willd.

Answer: Material source was added to the manuscript, they were obtained from heartwood.

Comment: Line 113: Chemical composition The full references for the TAPPI standards are missing from the reference list. The same applies to the ASTM standards.

Answer: References for TAPPI and ASTM were inserted in reference section.

Comment: Line 119: In what conditions were the samples stored, and for how long? What type of density have you measured- green, oven-dried, or basic density? Please provide the manufacturer, as well as the city and country of production, for the universal testing machine. Even though you mentioned that you followed a standard, it is important to provide the sample size, temperature, and relative humidity at which samples were acclimatized etc.

Answer: The samples were oven dried, the provider and country of the machine was added as well.

Comment: Line 133: Please mention the amount of samples taken for this analysis. Same comment for the equipment as mentioned above.

Answer: For this analysis in specific, we measured multiple times but in the manuscript the result is show for one sample.

Comment: Statistical analysis: This section is needed to describe the statistical differences of measured properties among three studied species. Otherwise, it is not scientific to say one species has higher let’s say cellulose than other. Tables 1 and 2 need statistical test data.

Answer: Thank you for the comment, the variability of the data really did create significant uncertainty in what was expected in a comparative analysis. To ensure a better analysis, the Weibull method was added with 95% confidence of all results, thus ensuring that the comparative evaluation was carried out.

Comment: Results: Is this section for results only, or does it include both results and discussion? The numbering of the sections is also incorrect, as it jumps from 3. Results to 5. Conclusion.

Answer: This section is for results and discussion. The section name was updated and the numbering as well.

Comment: Line 175, 232, 252, 270: Scientific names should be in italic. Please be consistent throughout the manuscript.

Answer: All the scientific names are now in italic.

Comment: Line 228: Please provide the full meaning of DTG once it appears first.

Answer: The full meaning of DTG was added.

Comment: Line 232-278: This description can be condensed by summarizing Figure 4 instead of describing it separately for each of the three species. Choose one key point and compare it across all three species. Please also support your statements with references from previous findings.

Answer: We chose to explain each wood species separately to clarify their individual behaviors. Additionally, more references have been added to this section.

Round 2

Reviewer 1 Report

Comments and Suggestions for Authors

I am very glad that the authors listened to my recommendations and corrected all the comments. I hope to see a study on the decomposition and oxidation kinetics of these woods, as well as an analysis of the outgoing gases. I recommend the article for acceptance.

Author Response

Comment: I am very glad that the authors listened to my recommendations and corrected all the comments. I hope to see a study on the decomposition and oxidation kinetics of these woods, as well as an analysis of the outgoing gases. I recommend the article for acceptance.

Response: We sincerely appreciate your positive feedback and valuable suggestions. We are pleased to know that the revisions have effectively addressed your comments. Your suggestion for this study  is very insightful and will be considered for future research. Thank you for recommending our paper for acceptance.

Reviewer 2 Report

Comments and Suggestions for Authors

Thanks for correcting your manuscript according to my comments. Please address the comment below:

2.6 Statistical analysis: Please provide statistical analysis description using this subheading. Description from Line 179-183 can be moved in this section. Also provide which software you have used for statistical analysis.

Line 322: The scientific name should be written in italics. Either include it in full or use the abbreviated form, such as 'TOC,' for consistency. Similar to line 284 and 304.

Line 358: Figure 6b.

Author Response

Comment: 2.6 Statistical analysis: Please provide statistical analysis description using this subheading. Description from Line 179-183 can be moved in this section. Also provide which software you have used for statistical analysis.

Response: We have added a new section as suggested by the reviewer. In this new section we have added the equations, parameters and software used to perform the statistical analyses.

Line 322: The scientific name should be written in italics. Either include it in full or use the abbreviated form, such as 'TOC,' for consistency. Similar to line 284 and 304.

Response: We changed the description of the types of wood according to the suggestion to maintain consistency. 

Line 358: Figure 6b.

Response: Thank you for the review, we have made the suggested corrections.
